# Antihistamines Potentiate Dexamethasone Anti-Inflammatory Effects. Impact on Glucocorticoid Receptor-Mediated Expression of Inflammation-Related Genes

**DOI:** 10.3390/cells10113026

**Published:** 2021-11-05

**Authors:** Carlos Daniel Zappia, Valeria Torralba-Agu, Emiliana Echeverria, Carlos P. Fitzsimons, Natalia Fernández, Federico Monczor

**Affiliations:** 1Facultad de Farmacia y Bioquímica, Universidad de Buenos Aires, Buenos Aires 1113, Argentina; danielzappia@hotmail.com (C.D.Z.); valeriat293@gmail.com (V.T.-A.); emilianaechev@gmail.com (E.E.); natycfernandez@gmail.com (N.F.); 2Consejo Nacional de Investigaciones Científicas y Técnicas (CONICET), Instituto de Investigaciones Farmacológicas (ININFA), Universidad de Buenos Aires, Buenos Aires 1113, Argentina; 3Center for Neuroscience, Faculty of Science, Swammerdam Institute for Life Sciences, University of Amsterdam, 1098 XH Amsterdam, The Netherlands; c.p.fitzsimons@uva.nl

**Keywords:** inflammation, dexamethasone, antihistamines, glucocorticoids, histamine

## Abstract

Antihistamines and glucocorticoids (GCs) are often used together in the clinic to treat several inflammation-related situations. Although there is no rationale for this association, clinical practice has assumed that, due to their concomitant anti-inflammatory effects, there should be an intrinsic benefit to their co-administration. In this work, we evaluated the effects of the co-treatment of several antihistamines on dexamethasone-induced glucocorticoid receptor transcriptional activity on the expression of various inflammation-related genes in A549 and U937 cell lines. Our results show that all antihistamines potentiate GCs’ anti-inflammatory effects, presenting ligand-, cell- and gene-dependent effects. Given that treatment with GCs has strong adverse effects, particularly on bone metabolism, we also examined the impact of antihistamine co-treatment on the expression of bone metabolism markers. Using MC3T3-E1 pre-osteoblastic cells, we observed that, though the antihistamine azelastine reduces the expression of dexamethasone-induced bone loss molecular markers, it potentiates osteoblast apoptosis. Our results suggest that the synergistic effect could contribute to reducing GC clinical doses, ineffective by itself but effective in combination with an antihistamine. This could result in a therapeutic advantage, as the addition of an antihistamine may reinforce the wanted effects of GCs, while related adverse effects could be diminished or at least mitigated. By modulating the patterns of gene activation/repression mediated by GR, antihistamines could enhance only the desired effects of GCs, allowing their effective dose to be reduced. Further research is needed to correctly determine the clinical scope, benefits, and potential risks of this therapeutic strategy.

## 1. Introduction

Inflammation is a physiological response aimed to fight a potential aggressor. Although originally beneficial, inflammation could become destructive when not controlled, leading to chronic inflammatory conditions such as asthma, chronic obstructive pulmonary disease, rheumatoid arthritis, inflammatory bowel disease and inflammation-related cancer and cardiovascular diseases. Chronic inflammatory diseases are the most significant cause of death in the world. The prevalence of maladies associated with chronic inflammation is going to increase incessantly for the next 30 years. Rand Corporation estimates that, in 2014, nearly 60% of Americans had at least one chronic condition, 42% had more than one and 12% of adults had 5 or more chronic conditions. Worldwide, three out of five deaths are due to chronic inflammatory diseases like stroke, chronic respiratory diseases, heart disorders, cancer, obesity, and diabetes [1,2].

Corticoids are one of the most used drugs in the world to treat numerous inflammatory and immune diseases [3]. These multifunctional hormones were first recognized as anti-inflammatory agents in the 1940s, and nowadays, their synthetic analogues are the most prescribed medications for this purpose [4]. Nevertheless, their associated adverse effects limit their use and propel both the optimization and search for new therapeutic strategies to combat those limitations.

To exert their anti-inflammatory effect, glucocorticoids bind to the glucocorticoid receptor (GR) and regulate gene expression positively or negatively by binding to promoter regions of target genes or by interaction with other transcription factors [5]. Recent works suggest rapid effects of the GR on inflammation that might not be mediated by changes in gene expression. However, it is not clear the role that these mechanisms have on its anti-inflammatory action [6]. Transrepression of proinflammatory genes mainly occurs through the interference of the GR with other proinflammatory transcription factors, such as NF-κB or AP-1, which results in a reduction of numerous pro-inflammatory mediators synthesis, such as IL-6, IL-8, TNF-α and IFN-γ, among others [6,7,8].

Although anti-inflammatory effects were originally associated with the repression of gene expression [9,10,11,12], and the clinical success of the GCs as anti-inflammatory drugs is historically related to its ability to inhibit pro-inflammatory transcription factors [3,6,11], transactivation of anti-inflammatory genes is necessary to exert a complete anti-inflammatory response [13,14,15]. Numerous genes are induced by GCs, among which are I-κB, GILZ, DUSP-1 (MKP-1), Annexin-1, and IL-10, [4,11]. MAPK phosphatase 1 (MKP-1 or DUSP-1) is one of the most potent anti-inflammatory proteins induced by the GR whose importance lays on the dephosphorylation of all members of the MAPK family that play important roles in the immune system [16]. For its part, GILZ (glucocorticoid-induced leucine zipper) is a gene with important anti-inflammatory and immunomodulatory properties whose induction is related to the effects of GCs on the activation of the MAPK cascades and the proinflammatory transcription factor NF-κB. A great number of reports show the particular importance of these two genes concerning GR’s anti-inflammatory effects [17,18].

Despite the undeniable beneficial effects of GCs, their use at high doses and for long periods could lead to severe adverse effects, especially those related to bone metabolism [19]. Osteoporosis is one of the most common and serious adverse effects which affects up to 50% of patients treated with GCs [20]. The mechanisms underlying this effect rely on two key proteins involved in bone resorption, osteoprotegerin (OPG) and the receptor activator of nuclear factor kappa-B ligand (RANKL). RANKL is expressed in the membrane of osteoblasts and sustains osteoclast differentiation, while OPG is an osteoclastogenesis inhibitor also expressed in osteoblasts that suppresses bone resorption [21]. Inasmuch, the ratio RANKL/OPG is considered a biomarker of bone resorption [22]. GCs commonly increase RANKL and reduce OPG expression, changing their ratio and, as a consequence, increasing osteoclast activity, which finally leads to osteoporosis [23]. Another common marker of bone formation and osteoblast function is the osteoblast-secreted protein osteocalcin (OC), which is known to be negatively regulated by GCs [22]. 

We have reported the existence of crosstalk between histamine H1 receptor (H1R) signalling and GR transcriptional activity and described the underlying molecular mechanism of this cross-regulation. According to our results, histamine (HA), acting on the H1R, potentiates dexamethasone-induced GR transcriptional activity. Furthermore, clinically relevant antihistamines also enhance dexamethasone’s response for GR-dependent transactivation and transrepression in a gene-specific way [24]. We discussed that the existence of cell types co-expressing H1R and GR suggests that our findings may have implications for regulation of inflammation in several systems, and the co-administration of corticoids and antihistamines could result in a reduction of the GC dose needed to achieve a therapeutic effect. As a proof-of-concept, we demonstrated an effective use for the treatment of asthma in an experimental murine model [25]. However, there is a need for a careful evaluation of the side effects of antihistamine and GC co-administration, because GCs’ undesired effects could also be enhanced by antihistamines [26].

In the present work, we evaluated in vitro the co-treatment of antihistamines and glucocorticoids at a molecular level, focusing on inflammation-related gene expression and corticoids’ detrimental effects. Given the widespread association of both drugs, their molecular interaction must be necessarily considered. Understanding the consequences of this interaction will provide a solid basis to optimize current therapies or develop novel strategies to treat inflammation-related conditions in an efficacious and safe approach.

## 2. Materials & Methods

### 2.1. Materials

RPMI 1640 and DMEM mediums, antibiotics, phosphate-buffered saline (PBS), phorbol 12-myristate 13-acetate (PMA), bovine serum albumin (BSA), ascorbic acid, β-glycerophosphate, dexamethasone, cetirizine, chlorpheniramine, azelastine and diphenhydramine were obtained from Sigma Chemical Company (St. Louis, MO, USA). Mepyramine maleate and trans-triprolidine were from Tocris Cookson Inc. (Ballwin, MO, USA). Fetal bovine serum (FBS) was purchased from Natocor (Córdoba, Argentina). All other chemicals were of analytical grade and obtained from standard sources.

### 2.2. Plasmid Constructions

pRSV-GR was cloned by Keith Yamamoto [27]. pCEFL-H1R was previously generated in our laboratory [24]. IL6-Luc was a gift from Karolien De Bosscher (VIB Department of Medical Protein Research, University of Gent, Ghent, Belgium). 

### 2.3. Cell Culture

HEK293T (human embryonic kidney, ATCC CRL-3216) and A549 (human pulmonary, ATCC CCL-185) cells were cultured in Dulbecco’s modified Eagle’s medium (DMEM). U937 (human promonocytic, ATCC CRL-1593.2) and MC3T3-E1 Subclone 4 (murine pre-osteoblastic, ATCC CRL-2593) cells were cultured in RPMI 1640 medium. All mediums were supplemented with 10% fetal calf serum and 5 μg/mL gentamicin, and cells were incubated at 37 °C in a humidified atmosphere containing 5% CO_2_.

To perform the different experiments, HEK293T and A549 cells were incubated with TNF-α, dexamethasone, mepyramine, trans-triprolidine, cetirizine, chlorpheniramine, diphenhydramine, and/or azelastine as indicated. U937 cells were treated with PMA for differentiation and then stimulated with LPS, dexamethasone, mepyramine, trans-triprolidine, cetirizine, chlorpheniramine, diphenhydramine, and/or azelastine as indicated. MC3T3-E1 cells were incubated with ascorbic acid and β-glycine for cell differentiation, and then stimulated with dexamethasone and/or azelastine as indicated.

#### Transfection and Reporter Gene Assays

HEK293T cells seeded on 24-well plates were co-transfected using the K2 Transfection System (Biontex, Munich, Germany) with the IL6-Luc luciferase reporter plasmid, pCEFL-H1R and pRSV-GR according to the manufacturer’s instructions. After 4 h, cells were seeded in 96-well plates, and 24 h later cells were starved overnight and then stimulated with diverse agents. After a kinetic assessment, luciferase activity was measured at the optimal time of 24 h later using the Steady-Glo Luciferase Assay System according to the manufacturer’s instructions (Promega Biosciences Inc., San Luis Obispo, CA, USA) using a FlexStation 3 Multi-Mode Microplate Reader (Molecular Devices, LLC, San José, CA, USA). Experimental reporter activity was normalized to control activity. No differences were observed in results normalized to renilla-luc or protein expression levels.

### 2.4. RT-PCR and Quantitative Real-Time PCR

Total RNA was isolated from A549, U937 or MC-3T3 cells using Quick-Zol reagent (Kalium Technologies, Bernal, Buenos Aires, Argentina) following the manufacturer’s instructions. Samples consisting of 500 ng of RNA were teated with 1 μL DNAse I (RNAse free), incubated for 20 min at 37 °C, and then incubated for 10 min at 37 °C with Stop Solution according to the manufacturer’s instructions (Ambion, Life Technologies, Grand Island, NY, USA). For the first-strand cDNA synthesis, 1 µg of total RNA was reverse-transcribed using the High-Capacity cDNA Reverse Transcription kit (AB) with random primers. Quantitative real-time PCR (qPCR) was performed in triplicate on the Rotor Gene Q cycler (Qiagen, Germantown, MD, USA) using the resulting cDNA, the HOT FIREPol EvaGreen qPCR Mix Plus (Solis Biodyne, Tartu, Estonia) for product detection, and the primers for human GILZ (glucocorticoid induced leucine zipper; NM_001015881.1); human MKP-1 (mitogen-activated protein kinase phosphatase 1; NM_004417.4); human IL-8 (Interleukin 8, NM_000584); human COX-2 (cyclooxygenase-2; NM_000963.1); human GMCSF (granulocyte-macrophage colony-stimulating factor; NM_000758); murine OC (osteocalcin BGLAP, NM_007541.3); murine OPG (osteoprotegerin, NM_008764.3); murine RANKL (receptor activator of nuclear factor kappa B ligand, NM_011613.3); human β-Actin (βAct, NM_001101.3,); and murine β-Actin (βAct, NM_007393.3), as described in Appendix A, presented in Appendix A. All primers were designed for the amplicon to include an intron to preclude DNA amplification. The cDNA was amplified by 45 cycles of denaturing (10 s at 95 °C), annealing (10 s at 60 °C), and extension (10 s at 72 °C) steps. To verify that the primer pairs used yielded single PCR products, a dissociation protocol was added after thermocycling, determining dissociation of the PCR products from 65 to 95 °C for 15 s. Finally, a cooling step was set for 20 s at 40 °C. 

To estimate the efficiency of the amplification reaction, serial half logarithm unit dilutions of cDNA from the A549 or U937 cells were used, and standard curves were generated. The linear slope of the standard curve for each primer pair was estimated using GraphPad Prism 6 software and the efficiency was calculated based on the following Equation (1).
(1)$$Efficiency=10^{-\left(1/slope\right)}$$

Additionally, the -RT samples and a water template were included in the analysis to confirm the absence of any residual DNA or contamination. All cDNA samples were analyzed in triplicates. Finally, the following Equation (2) was used to calculate the fold induction of gene expression using the double-ΔCT method [28].
(2)$$\begin{array}{c} Fold\;change=2^{-\Delta \Delta CT}=[(CT\;target\;gene-CT\;reference\;gene)\;experimental\;sample-\\ \left(CT\;target\;gene-CT\;reference\;gene)\;control\;sample\right] \end{array}$$

### 2.5. MTS Assay

Cell proliferation was determined by a colourimetric assay using CellTiter 96 AQueous Non-Radioactive Cell Proliferation Assay (Promega, Madison, WI, USA) according to the manufacturer’s instructions. For the MTS assay, cells growing in the exponential phase were seeded at 2.0 × 10^4^ cells/well in a 96-well plate, incubated in an atmosphere of 5% CO_2_ at 37 °C and exposed to different stimuli. After incubation, 20 μL of MTS was added to each well and further incubated for 2 h at 37 °C. The absorbance was measured at 490 nm using the FlexStation 3 microplate reader (Molecular Devices LLC. San José, CA, USA).

### 2.6. Data Analysis

Statistical analysis was performed with GraphPad Prism 6.0 (GraphPad Software for Science, San Diego, CA, USA). Results are expressed as mean ± SEM. Parametric statistical analysis was performed using one- or two-way ANOVA, followed by Bonferroni post hoc multiple comparisons. Differences were considered significant at *p* < 0.05. Statistical significance was denoted using letters above bars. Means indicated with a common letter are not significantly different. Likewise, means not sharing any letter are significantly different.

## 3. Results

### 3.1. Antihistamines Enhance Dexamethasone-Mediated Inhibition of NF-κB Activity Induced by TNF-α

We have previously shown that the antihistamines mepyramine (MEP) and azelastine (AZE) potentiate dexamethasone (DEX) inhibition of NF-κB activity in vitro on an IL-6 promoter-driven luciferase reporter [24,25]. To extend this study we evaluated the effect of the different antihistamines mepyramine (MEP), trans-triprolidine (TRIP), cetirizine (CET), chlorpheniramine (CHLOR) and diphenhydramine (DIPH) using HEK-293T cells co-transfected with plasmids coding for GR, H1R and the aforementioned reporter. Luciferase activity was induced by pre-exposing cells to 2000 UI/mL TNF-α, which was reduced in the presence of 1 nM DEX. A total of 10 min pre-treatment with 10 μM of all antihistamines, except CHLOR, enhanced the inhibition induced by DEX 3 to 4 times (Figure 1).

### 3.2. Modulation of Endogenous Pro-Inflammatory Gene Expression

Since transrepression of pro-inflammatory genes is the primary mechanism by which corticoids exert their anti-inflammatory effects, we evaluated if the results obtained in the transrepression of the luciferase-reporter system were replicated in pathophysiologically relevant contexts. We studied the modulation of endogenous pro-inflammatory genes activated by NF-κB and transrepressed by the GR in A549 alveolar epithelial cells stimulated with the pro-inflammatory cytokine TNF-α, chosen as a model of alveolar reactivity and pulmonary inflammation [24], and in U937 promonocytic cells, first differentiated to macrophages with phorbol 12-myristate 13-acetate (PMA) and then stimulated with the bacterial endotoxin lipopolysaccharide (LPS), derived from the immune system and chosen as a suitable model to evaluate the modulation of pro-inflammatory genes [29,30,31,32,33]. We measured the expression of interleukin 8 (IL-8), representative of the chemokines induced by NF-κB and implicated in several pulmonary diseases; cyclooxygenase 2 (COX-2), characteristic of the enzymes induced by NF-κB and the molecular target of non-steroidal anti-inflammatory drugs; and the granulocyte-macrophage colony-stimulating factor (GM-CSF), representing the cytokines induced by NF-κB and associated with the GR’s histone acetylation mechanism of action. In A549 cells, treatment with 2000 UI/mL TNF-α induced the expression of all three genes, which was reduced when cells were treated with 1 nM DEX. Incubation with 10 nM DEX induced a decrease in gene expression below basal levels that prevented observation of the effects of the antihistamines (Appendix A). Pre-treatment with 10 µM CHLOR, DIPH and AZE enhanced the DEX effect for IL-8 and COX-2, while 10 µM MEP potentiated the DEX effect only for IL-8. For their part, TRIP, CET and CHLOR also potentiate the DEX effect on GM-CSF expression (Figure 2A–C respectively). In U937 cells, treatment with 1 μg/μL LPS showed an increase in the remarkably low basal mRNA levels of IL-8, COX-2 and GM-CSF. In this system, treatment with 1 nM DEX resulted in an inhibition of the LPS effect that was enhanced by pre-treatment with 10 µM TRIP, CET, CHLOR, DIPH and AZE for all genes evaluated, while 10 µM MEP enhanced the DEX effect only for COX-2 and GM-CSF (Figure 2D–F respectively).

### 3.3. Modulation of Endogenous Anti-Inflammatory Gene Expression

In addition to transrepression, several anti-inflammatory genes are transactivated by the GR as an essential part of its anti-inflammatory mechanism of action. As was mentioned, GILZ and MKP-1 are of particular importance. In both U937 and A549 cell lines, treatment with DEX induced GILZ and MKP-1 expression. Incubations were performed using 10 nM DEX, which induced a 3-to-4-fold rise in gene expression, except for GILZ in A549, where 1 nM DEX stimulus was used because a concentration increase induced a 12-fold elevation that could prevent observing the effects of antihistamines (Appendix A). This effect was significantly enhanced in A549 cells pre-treated with TRIP, CET, CHLOR, DIPH and AZE for GILZ and with TRIP, CET, CHLOR and AZE for MKP-1 (Figure 3A,B). In U937 cells, pre-treatment with MEP, CET and AZE showed an enhancement of GILZ induction by DEX, while pre-treatment with CHLOR and DIPH showed this same effect for DEX-induced MKP-1 expression (Figure 3C,D).

### 3.4. Modulation of Bone Metabolism Markers Expression

The observation that antihistamines potentiate GR-mediated anti-inflammatory effects may have a beneficial impact on clinical treatments. However, it is crucial to evaluate the effects that they could have on GCs’ undesired adverse effects. For this study, we used the MC3T3-E1 pre-osteoblastic cell line (widely used as an in vitro osteoblast model to study the maturation of pre-osteoblastic cells into a matrix mineralizing osteoblast [34]) to investigate whether antihistamines could modulate corticoids’ effects on OPG, RANKL and OC gene expression. Cells were differentiated to osteoblasts with ascorbic acid (AA) and β-glycerophosphate (β-gly), which increased OPG, OC and reduced RANKL gene expression, resulting in a suitable model to evaluate corticoids’ well-known effects on their expression (Figure 4).

In this system, we evaluated the effect of the antihistamine AZE, as we have recently reported this ligand was able to increase DEX’s efficacy in a murine asthma model [25]. Our results showed that treatment of the cells with 0.1 nM DEX reduced OPG and increased RANKL expression and that both effects were reversed by pre-treatment with 10 µM AZE (Figure 5A). This result was reflected in the RANKL/OPG ratio, which is a marker of bone impairment and which was reduced when cells were co-treated with DEX and AZE (Figure 5B). Additionally, treatment with 1 nM DEX reduced OC gene expression, an effect that was also reversed when cells were pre-treated with AZE (Figure 5C). All these results show that the deleterious effects induced by DEX on the osteoblasts markers evaluated can be mitigated in the presence of AZE, suggesting that co-treatment might be a secure therapeutic strategy in terms of bone impairment.

### 3.5. Effect of AZE on Differentiated MC3T3-E1 Cell Proliferation

To assess the effect of AZE on AA and β-Gly-differentiated MC3T3-E1 cell proliferation, we measured cell viability by the MTS metabolic assay. We first performed a kinetic assay to obtain the optimal conditions for DEX-mediated effects. Treatment with 1 µM DEX inhibited cell proliferation, and this effect was at its maximum at 96 h (Figure 6A). At this experimental time, DEX-mediated inhibition of cell proliferation was concentration-dependent, reaching its maximal effect at 1 µM (Figure 6B). Pre-treatment with 10 µM AZE enhanced the inhibition of cell proliferation to a constant value for all the concentrations of DEX tested, suggesting that the effect might be saturated. Thus, we evaluated a lower concentration of AZE. Pre-treatment of cells with 1 µM AZE potentiated DEX-mediated inhibition of cell proliferation, and this effect was evidenced by a left-shift of the concentration-response curve (−8.49 ± 0.23 vs. −9.38 ± 0.20; *p* < 0.05, extra sum-of-squares F test) (Figure 6C). Interestingly, 10 µM AZE had a deleterious effect greater than that induced by the highest DEX concentration, which almost disappeared under the ten-fold reduction of AZE concentration. All of these results suggest that co-treatment with AZE and DEX might be risky in terms of bone impairment and seems to oppose the previous results. These discrepancies will be discussed in the following section.

## 4. Discussion

The main conclusion of the present work is that, in general, the antihistamines evaluated enhanced DEX-induced anti-inflammatory effects in vitro. The results obtained for the antihistamines’ potentiation of several DEX-induced GR-regulated endogenous inflammation-related genes in two cell models support this assertion (Table 1). The cell type, gene and ligand differences observed preclude a simple generalization of the modulatory effects and reveal the complexity of the phenomenon studied. The more effective ligands were CHLOR, DIPH and AZE, but to extend the analysis, pharmacological differences between the antihistamines evaluated and the importance of cofactors and response elements on the GR transcriptional mechanism of action will be considered.

Although all antihistamines share their pharmacological activity, they can differ in their molecular mechanisms of action. For instance, it has been documented that MEP binds preferentially to the inactive, but G-protein coupled, state of the H1R, while TRIP binds preferentially to the inactive but uncoupled state of the receptor [35]. Expanding this analysis to all the antihistamines evaluated herein, their affinities and efficacies on different responses has been reported [36,37]. Although all of them behave as inverse agonists, reducing H1 receptor constitutive activity concerning IP levels, there are significant differences in the different parameters evaluated. For example, CET presents the maximum efficacy to reduce IP levels, but the lowest efficacy to reduce NF-κB activity, while inversely, MEP presents the maximum efficacy to diminish NF-κB activity but a moderate efficacy to decrease IPs levels. Moreover, ligand bias of GPCR signaling is a well-recognized general phenomenon that was already described for antihistamines. Ligand bias acknowledges that ligand efficacy is a pathway-dependent property. It was recently reported that certain antihistamines, besides their negative efficacy to modulate IP levels, mimic histamine effects by behaving as agonists, increasing ERK phosphorylation. In this regard, while TRIP and DIPH act as full agonists, CHLOR functions as a partial agonist [38]. These differences suggest that the molecular mechanisms by which antihistamines exert their effects are different, as was documented for MEP and TRIP (for a review on inverse agonists mechanisms of action please refer to Monczor, 2013 [39]). Concerning GR transcriptional activity modulation, the antihistamines MEP and TRIP increase GR transactivation of the reporter TAT3-LUC with similar potency and efficacy [24]. However, when they were evaluated by their ability to modulate GR transrepression of the reporter IL6-LUC, they showed differences in their efficacies, suggesting that the molecular mechanism involved in transactivation could be different from that involved in transrepression. As discussed, these differences could rely on the mechanisms by which the antihistamines act on the H1 receptor. The results showed herein prove that, except for COX-2 and GMSCF in U937 cells, when a given gene and cell type are considered, antihistamines differ in their efficacies for the modulation of the DEX response. Further research on antihistamines signalling is needed to get a better understanding of the influence of ligands’ mechanism of action on the modulation of GR activity.

In addition, differences in the modulation of the expression of different DEX-induced genes were observed for the same histaminergic ligand. This suggests that antihistamines’ effect depends on other factors, such as the pool of cofactors (co-activators and co-repressors) expressed by the cells (cell dependence) and each gene response element (gene dependence). GR activation and response is not a simple binary process. On the contrary, numerous events take place between ligand-receptor-DNA interaction and gene modulation [40], including the binding to cofactors whose differential expression and recruitment lead to cell type-specific gene expression patterns [9,41,42,43,44]. GR, DNA and cofactors form functional regulatory complexes that remodel chromatin and modify the activity of the transcription machinery [45]. The transcriptional activity of the regulatory complexes in a given response element will be determined by three factors: the sequence of the response element (GRE sites are not a unique sequence but composite elements formed by a family of diverse sequences), its availability to bind the GR and the nature of the cofactors present in the cell [44]. It has been shown that these processes interact cooperatively [5,46]. These considerations may account for the differences observed between antihistamines’ effect on DEX transactivation of GILZ and MKP-1 gene expression. While DEX induced the expression of GILZ and MKP-1 in A549 and U937 cells, pre-treatment with TRIP, CET, CHLOR, DIPH and AZE enhanced GILZ induction in A549 cells, but only MEP, CET and DIPH did the same in U937 cells. A similar analysis for MKP-1 shows that pre-treatment with TRIP, CET, CHLOR and AZE enhanced its DEX-mediated induction only in A549 cells, while only DIPH and CHLOR did the same in U937 cells, reflecting cell-type differences. If the same cell line is considered, pre-treatment with TRIP, CET, CHLOR, and AZE enhanced GILZ and MKP-1 induction in A549 cells, while DIPH had an effect only for GILZ. In U937 cells, pre-treatment with MEP, CET and DIPH enhanced GILZ induction while CHLOR and DIPH enhanced MKP-1 induction. These results suggest the importance of differences in GRE sequences.

For its part, gene transrepression mediated by the antagonism of NF-κB activity can be achieved in different ways, but the mechanisms are still controversial [47]. Regarding cell-dependent effects, it has been described that GCs inhibit IL-8 levels by a mechanism that involves histone acetylation in U937 cells [48], while a histone-independent mechanism involving the phosphorylation of the RNA pol II C-terminal domain was described in A549 [49]. Considering our results, these differences in the mechanism of action could justify why MEP, TRIP and CET have different effects on the modulation of DEX-mediated inhibition of IL-8 gene expression in U937 and A549 cells.

Concerning gene specificity, it has been described that GCs inhibit GMCSF expression by a histone-dependent mechanism in A549 cells [50], while for COX-2 it has been proven that DEX can inhibit its expression by direct interaction of the GR with NF-κB [51], or by increasing its mRNA metabolism [52]. Again, these differences could explain the ability of TRIP, CET, DIPH and AZE to differentially modulate DEX-induced repression of GMCSF and COX-2 gene expression in A549 cells.

In conclusion, ligands, cofactors, GR post-translational modifications and specific events at each promoter govern GR molecular mechanisms of action and ultimately determine gene and cell specificity of its transcriptional activity. In our case, the different antihistamines evaluated and their specific H1R-mediated mechanism of action, the particular modifications they can induce on GR transcriptional activity, the pool of cofactors expressed by each cell line and the specific events that can take place at promoters of the different genes evaluated could explain the ligand, gene and cell specificity observed for the modulation of GR-regulated endogenous gene expression by each antihistamine. A limitation to our study is the lack of a discovery-oriented genome-wide transcriptome analysis using RNA-Seq. It would be of interest to unveil unsuspected gene expression regulation. 

Regardless of this molecular disquisition, the finding that most of the antihistamines enhanced DEX-mediated effects both for anti-inflammatory gene transactivation and for pro-inflammatory gene transrepression is pharmacologically relevant. Considering the clinical significance of corticoid and antihistamine co-treatment, our results give rationale to a very commonly-used drug association as well as help to design new therapeutic strategies to treat inflammatory conditions or diseases. Co-treatment might allow the doses of corticoids needed to reach their anti-inflammatory effect to be reduced, which would represent a therapeutic advantage [53]. Supporting this hypothesis, we have reported that the antihistamine AZE was able to enhance DEX-mediated effects in a murine asthma model, providing new insights into the potential benefits of the combination for the management of asthma [25].

Nevertheless, corticoids’ associated adverse effects might be enhanced as well, considering that they share the same transcriptional mechanisms as the anti-inflammatory effects. GC-induced adverse effects include osteoporosis, hypothalamus-pituitary-adrenal (HPA) axis suppression, growth retardation, cataract formation, skin thinning and bruising. Since the most serious and debilitating of these is GC-induced osteoporosis (GIOP) [54], we evaluated the consequences of co-treatment with AZE and DEX on bone metabolism in vitro.

Bone loss in the chronic state of GIOP is mostly attributable to the decreased bone formation by osteoblasts, impaired osteoblast cell replication, diminished osteoblast differentiation and function, and accelerated osteoblast and osteocyte apoptosis. GCs also had direct effects on osteoclasts, stimulating bone resorption and osteocytes as well [55].

We determined the effects of AZE and DEX co-treatment in the differentiated MC3T3-E1 pre-osteoblastic cell line. Our results show that treatment of osteoblasts with DEX increases the RANKL/OPG ratio, and the addition of AZE restores this parameter to basal levels. Interestingly, AZE alone was able to reduce the RANKL/OPG ratio, suggesting that this effect might be counteracting the rise induced by DEX. The role of histamine on bone metabolism has been investigated in recent years. HA promotes osteoclastogenesis by targeting osteoblasts and increasing its RANKL/OPG ratio, which in turn increases osteoclast differentiation and activity [56]. Ikawa and coworkers also reported the H1R-mediated induction of RANKL by HA in MC3T3-E1 cells, and they noted “a very little reduction” mediated by mepyramine [57]. In addition, we found that co-treatment with AZE and DEX also restored the DEX-reduced OC levels, reflecting the effect of the co-treatment on bone formation and osteoblast activity. Taken together, our results show that DEX-related adverse effects were not only not potentiated but reversed by the addition of AZE. However, when osteoblast viability was evaluated, AZE enhanced DEX-induced inhibition of MC3T3-E1 cell proliferation. As with the RANKL/OPG ratio, AZE alone showed an effect by itself, in this case inhibiting cell proliferation. 

Clinical studies have described the protective effect of antihistamines against bone loss. Patients with osteoporosis taking antihistamines as an antiallergic treatment have shown higher bone density [58], suggesting that HA blocking by these drugs may protect against bone osteoporosis. This is in line with the role that HA has on bone resorption [59]. Since the bibliography refers to different antihistamines from those assayed herein, and we showed that not all antihistamines share all pharmacological/clinical properties, it is not possible to generalize antihistamine biological effects. It remains to address the global effect that AZE and DEX co-treatment could have in vivo on bone metabolism. In our case, this will depend on the contribution of each process to GC-induced osteoporosis.

Glucocorticoids are used to treat a wide variety of inflammatory conditions, such as asthma or rheumatoid arthritis, among others, and in some cases are irreplaceable. However, treatment with corticoids has strong adverse effects. Several strategies pursue the development of GC-based therapies with reduced adverse effects, involving chemical optimization of physiological corticoids or the development of selective glucocorticoid receptor agonists and modulators (SEGRAs and SEGRMs). The limited clinical success of these strategies reflects the under-appreciation of the complexity of GR transcriptional activity by opposing GR-mediated mechanisms in immune suppression vs. side effects. On the contrary, our results manifest how intricate the GR response and its modulation is, involving ligand, cell and gene specificity. By modulating the patterns of gene activation/repression mediated by the GR, antihistamines could enhance GCs’ therapeutic effects, allowing their effective dose to be reduced. Further research is needed to correctly determine the clinical scope, benefits, and potential risks of this therapeutic strategy.

## Figures and Tables

**Figure 1 cells-10-03026-f001:**
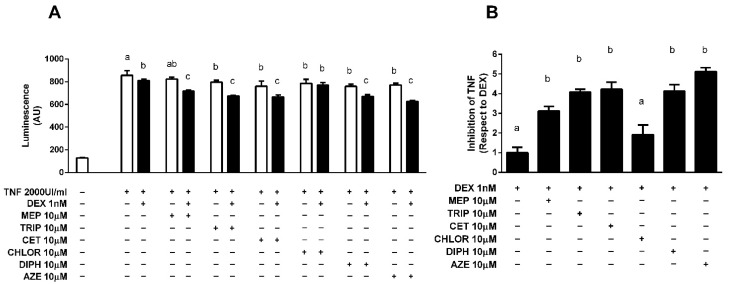
Antihistamines potentiate dexamethasone-induced GR transrepression of a gene-reporter system. (**A**) HEK-293T cells were co-transfected with IL6-Luc, GR and H1R coding constructs and were incubated with 2000 UI/mL TNF-α for 4 h and exposed to 10 μM antihistamines for 10 min and to 1 nM dexamethasone (DEX) for 24 h (filled black bars) or not (empty white bars), as indicated. (**B**) Data express inhibition of the TNF-α-induced response as fold of the DEX effect. Luciferase activity was determined as described in the methodology section. Results are mean +/− SD of five independent experiments performed in triplicates. Statistical significance was denoted using letters above bars. Means indicated with a common letter are not significantly different. Likewise, means not sharing any letter are significantly different.

**Figure 2 cells-10-03026-f002:**
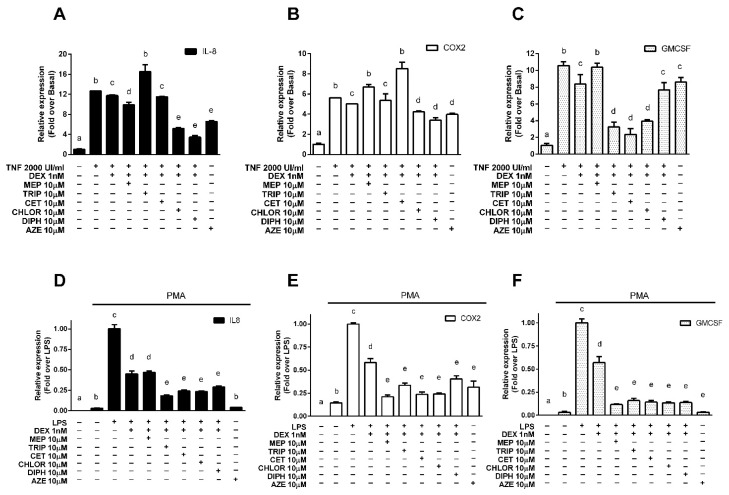
Antihistamines potentiate dexamethasone-induced transrepression of endogenous pro-inflammatory genes. (**A**–**C**) A549 cells were incubated with 2000 UI/mL TNFα for 4 h, exposed for 10 min to 10 μM mepyramine (MEP), 10 μM trans-triprolidine (TRIP), 10 μM cetirizine (CET), 10 μM chlorpheniramine (CHLOR), 10 μM diphenhydramine (DIPH), or 10 μM azelastine (AZE) as indicated, and then treated with 1 nM dexamethasone (DEX) for 3 h. (**D**–**F**) U937 cells were differentiated to macrophages with 100 nM PMA for 48 h and then stimulated with 1μg/μL LPS for 4 h. They were then treated for 10 min with 10 μM mepyramine (MEP), 10 μM trans-triprolidine (TRIP), 10 μM cetirizine (CET), 10 μM chlorpheniramine (CHLOR), 10 μM diphenhydramine (DIPH), or 10 μM azelastine (AZE) as indicated, and then exposed to 1 nM dexamethasone (DEX) for 3 h. IL-8, COX-2, and GMCSF mRNA levels were quantified by qPCR, as described in the methods section. Results are mean +/− SD of at least three independent experiments performed in triplicates. Statistical significance was denoted using letters above bars. Means indicated with a common letter are not significantly different. Likewise, means not sharing any letter are significantly different.

**Figure 3 cells-10-03026-f003:**
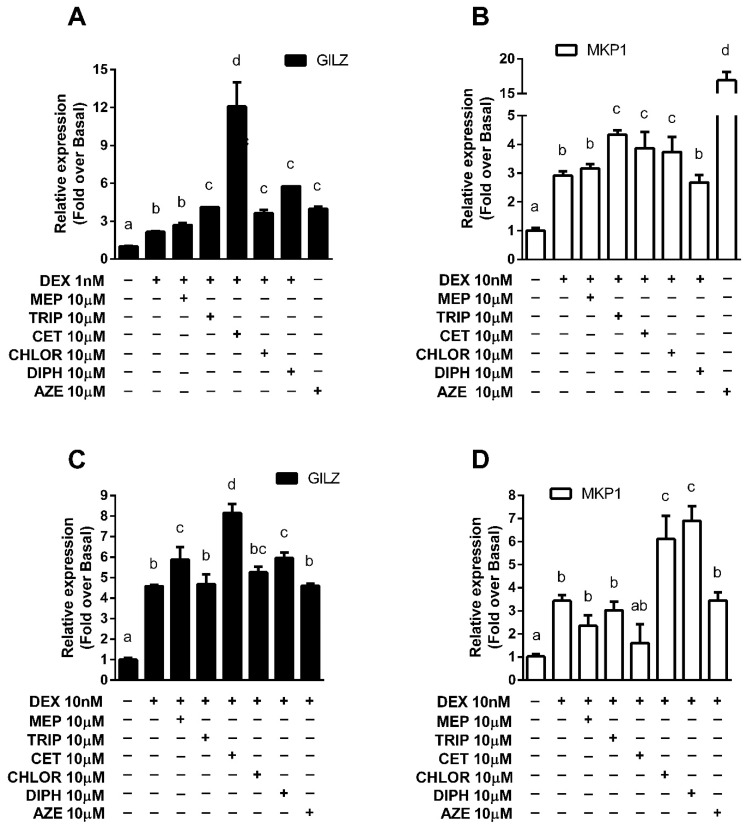
Antihistamines potentiate the dexamethasone-induced expression of endogenous anti-inflammatory genes. (**A**,**B**) A549 cells or (**C**,**D**) U937 cells were exposed for 10 min to 10 μM mepyramine (MEP), 10 μM trans-triprolidine (TRIP), 10 μM cetirizine (CET), 10 μM chlorpheniramine (CHLOR), 10 μM diphenhydramine (DIPH), or 10 μM azelastine (AZE) as indicated, and then treated with dexamethasone (DEX) for 3 h, as indicated. GILZ and MKP-1 mRNA levels were quantified by qPCR as described in the methods section. Results are mean +/− SD of at least three independent experiments performed in triplicates. Statistical significance was denoted using letters above bars. Means indicated with a common letter are not significantly different. Likewise, means not sharing any letter are significantly different.

**Figure 4 cells-10-03026-f004:**
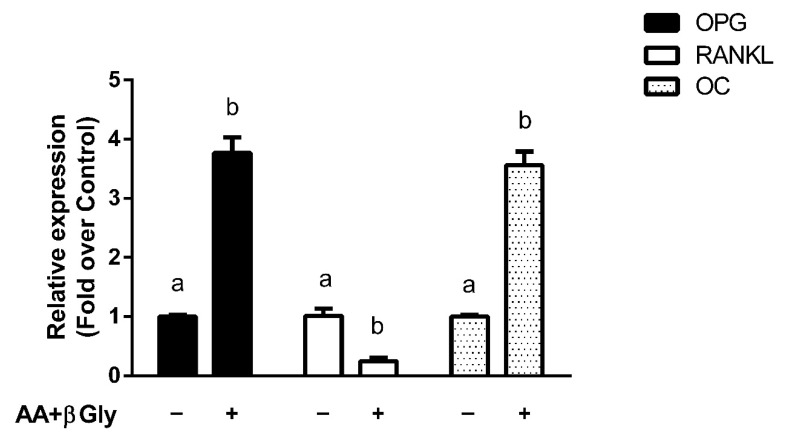
Differentiation of pre-osteoblastic cells induces changes in the expression of bone metabolism marker genes. MC3T3-E1 pre-osteoblastic cells were differentiated to osteoblasts with 50 μg/mL ascorbic acid (AA) and 10 mM β-glycerophosphate (β-gly) for 14 days. Osteoprotegerin (OPG), receptor activator of nuclear factor kappa-B ligand (RANKL) and osteocalcin (OC) mRNA levels were quantified by qPCR as described in the methods section. Results are mean +/− SD of at least three independent experiments performed in triplicates. Statistical significance was denoted using letters above bars. Means indicated with a common letter are not significantly different. Likewise, means not sharing any letter are significantly different.

**Figure 5 cells-10-03026-f005:**
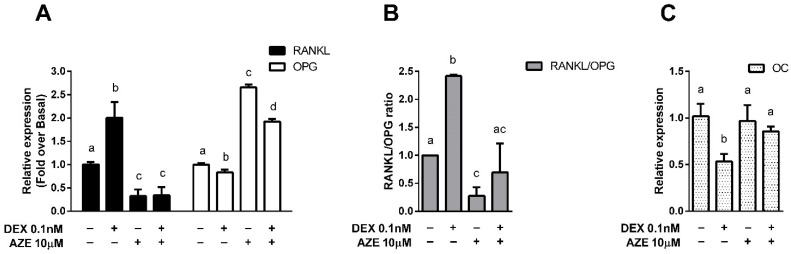
Azelastine counteracts dexamethasone-induced modulation of bone metabolism marker genes. MC3T3-E1 pre-osteoblastic cells differentiated to osteoblasts were exposed for 10 min to 10 μM azelastine (AZE) and then treated with 0.1 nM dexamethasone (DEX) for 3 h. (**A**,**B**) Receptor activator of nuclear factor kappa-B ligand (RANKL), osteoprotegerin (OPG), and (**C**) osteocalcin (OC) mRNA levels were quantified by qPCR as described in the methods section. Results are mean +/− SD of at least three independent experiments performed in triplicates. Statistical significance was denoted using letters above bars. Means indicated with a common letter are not significantly different. Likewise, means not sharing any letter are significantly different.

**Figure 6 cells-10-03026-f006:**
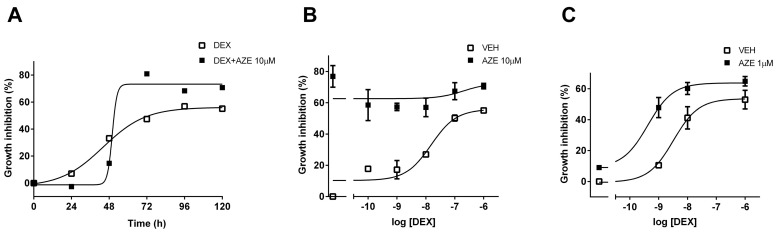
Azelastine potentiates dexamethasone-induced osteoblastic cell growth inhibition. (**A**) MC3T3-E1 pre-osteoblastic cells differentiated to osteoblasts were exposed to 10 μM azelastine (AZE) and treated with 1 nM dexamethasone (DEX) for different time periods. Maximal growth inhibition was achieved at 96 h. (**B**) Cells were treated with 10 μM azelastine (AZE) and then subjected to different concentrations of dexamethasone (DEX) for 96 h. (**C**) Cells were treated with 1 μM azelastine (AZE) and then subjected to different concentrations of dexamethasone for 96 h. Cell viability was measured by the MTS metabolic assay, as indicated in the methodology section. Results are mean +/− SD of at least three independent experiments performed in triplicates.

**Table 1 cells-10-03026-t001:** Effect of antihistamine on the dexamethasone-induced expression of inflammation-related genes.

	A549					U937				
	IL-8	COX2	GMCSF	GILZ	MKP1	IL-8	COX2	GMCSF	GILZ	MKP1
Mepyramine	3.12 ± 0.95 *	−2.71 ± 1.05 *	0.08 ± 0.38 *	1.26 ± 0.11	1.09 ± 0.08	0.97 ± 0.06	1.88 ± 0.09 *	2.05 ± 0.03 *	1.28 ± 0.23 *	0.68 ± 0.30
Trans-triprolidine	−4.41 ± 2.71 *	0.71 ± 2.96	3.32 ± 0.37 *	1.91 ± 0.01 *	1.49 ± 0.09 *	1.48 ± 0.04 *	1.59 ± 0.10 *	1.95 ± 0.10 *	1.02 ± 0.18	0.88 ± 0.24
Cetirizine	1.28 ± 0.07	−7.54 ± 4.00 *	3.73 ± 0.44 *	5.63 ± 1.26 *	1.33 ± 0.33 *	1.37 ± 0.03 *	1.82 ± 0.11 *	1.99 ± 0.08 *	1.78 ± 0.17 *	0.47 ± 0.57
Chlorpheniramine	8.47 ± 0.29 *	3.60 ± 0.39 *	3.01 ± 0.12 *	1.69 ± 0.23 *	1.28 ± 0.31 *	1.39 ± 0.03 *	1.81 ± 0.04 *	2.00 ± 0.04 *	1.15 ± 0.10	1.78 ± 0.41 *
Diphenhydramine	10.47 ± 0.64 *	5.69 ± 1.03 *	1.31 ± 0.55	2.69 ± 0.01 *	0.92 ± 0.15	1.28 ± 0.03 *	1.42 ± 0.14 *	2.00 ± 0.05 *	1.30 ± 0.10 *	2.01 ± 0.26 *
Azelastine	6.90 ± 0.34 *	2.83 ± 2.47 *	0.89 ± 0.37	1.85 ± 0.15 *	5.81 ± 0.73 *	1.74 ± 0.01 *	1.63 ± 0.27 *	2.25 ± 0.03 *	1.01 ± 0.04	1.01 ± 0.15

Values express the effect of each antihistamine with respect to the effect of DEX on the repression of IL-8, COX2 and GMCSF gene expression induced by TNF-α in A549 cells or LPS in PMA-differentiated U937 cells, or the induction of GILZ and MKP1 gene expression in A549 or U937 cells. Values > 1 indicate DEX potentiation, while values < 1 indicate DEX inhibition. * denotes statistically significant differences with respect to DEX at *p* < 0.05.

## Data Availability

Data available on request.

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
