# Peer review of "Antihistamines Potentiate Dexamethasone Anti-Inflammatory Effects. Impact on Glucocorticoid Receptor-Mediated Expression of Inflammation-Related Genes"

_cells, 2021, doi:10.3390/cells10113026_

Round 1

Reviewer 1 Report

The paper by Zappia and collaborators deals with the analysis of the possible synergistic effect due to the treatment with antihistamines and dexamethasone on the anti-inflammatory potential of the latter. The authors also analyzed the impact (positive or negative) of antihistamines on the harmful effects of corticoids’ treatment.

The results are clear and pharmacologically relevant with regards to the positive effect of treatment of antihistamines and DEX on the modulation of gene transcription of pro-inflammatory and anti-inflammatory markers.

However, since this crosstalk was already suggested by previous papers of these authors, different experimental approaches, such as RNAseq analysis, could help to not only confirm the crosstalk but also to find unknown interactions.

Moreover, the effects on bone metabolism are conflicting and difficult to understand. The authors had addressed this last aspect by taking into consideration many different antihistamine molecules, with diverse mechanisms of action. This prevents a deepening of the study and makes the discussion complex but not conclusive.

Furthermore, it is not explained how the concentrations of DEX  were chosen and why they are different in some experiments.

Minor

  • The authors must explain the choice of cell lines used (not always explained).
  • After RNA extraction no DNAse treatment was performed. Are all the primer pairs selected in different exons? If not, a possible amplification of genomic DNA cannot be ruled out.
  • A table showing the primers and their characteristics is certainly preferable to a list like the one reported in the paragraph: "RT-PCR and Quantitative real-time PCR ".
  • The DEX concentration is sometimes reported different in the graphs and in the figure legends (or in the text).
  • Quality of table 1 could be improved.

Author Response

Point-by point discussion of the reviewer’s 1 comments:

The paper by Zappia and collaborators deals with the analysis of the possible synergistic effect due to the treatment with antihistamines and dexamethasone on the anti-inflammatory potential of the latter. The authors also analyzed the impact (positive or negative) of antihistamines on the harmful effects of corticoids’ treatment.

The results are clear and pharmacologically relevant with regards to the positive effect of treatment of antihistamines and DEX on the modulation of gene transcription of pro-inflammatory and anti-inflammatory markers.

 However, since this crosstalk was already suggested by previous papers of these authors, different experimental approaches, such as RNAseq analysis, could help to not only confirm the crosstalk but also to find unknown interactions.

We agree with the reviewer that a discovery-oriented Genome-wide transcriptome analysis using RNA-Seq would be of interest to unveil unsuspected gene expression regulation. As we have indicated in our answer to the Academic Editor’s comments, this experiment is beyond our current possibilities. However, at this point, we would prefer to focus on a more hypothesis-driven set of experiments, rather than showing the effects on many genes that may or may not be of immediate interest, considering the anti-inflammatory effects of glucocorticoids and antihistamines. Acknowledging the reviewer’s criticism, we have included a sentence in the discussion point (page 13, lines 481 to 484) indicating this as a limitation of our study and how we would expect a Genome-wide transcriptome analysis using RNA-Seq to contribute to our work in the future.

Moreover, the effects on bone metabolism are conflicting and difficult to understand. The authors had addressed this last aspect by taking into consideration many different antihistamine molecules, with diverse mechanisms of action. This prevents a deepening of the study and makes the discussion complex but not conclusive.

We understand the point made by the reviewer regarding our results obtained in bone metabolism. In fact, we discussed our results considering the diversity of effects of several antihistamines on bone cells. We believe it is relevant for the discussion of our results to point out that histamine has a deleterious effect on the same cell line we use (MC3T3-E1), partially reverted by mepyramine, and that clinical studies found a protective effect of antihistamines against bone mass loss in patients with osteoporosis. However, it is true that in the cited studies, the reported antihistamines were different from those used in our work, opening the door to the possibility that different ligands, although sharing histamine blocking on H1R as the primary mechanism of action, present different effects on bone cells. This was added to the new version of the discussion. Please see page 14, lines 526 to 529.

Furthermore, it is not explained how the concentrations of DEX  were chosen and why they are different in some experiments.

Dexamethasone concentrations were chosen in accordance with their effect on gene expression, shown in a new Supplementary Section. Usually, increasing DEX concentrations resulted in a larger effect on gene expression, which would then prevent the effect of the antihistamine. For example, 10 nM DEX diminished TNF or PMA-stimulated pro-inflammatory gene expression levels below basal expression, and in A549 cells, 10 nM DEX induced GILZ expression to a level that precluded antihistamines to potentiate this effect, likely by a saturation effect. To help clarify this point, these results are now shown in the Supplementary Section (Figures S1 to S4) and discussed in the main text. Please refer to page 6 lines 267 to 269 and page 7 lines 296 to 300.

Minor

  • The authors must explain the choice of cell lines used (not always explained).

 We thank the reviewer for requesting this clarification. We choose the cell lines based on their characteristics and their suitability for the experiments proposed. A549 cells are a model of alveolar reactivity and pulmonary inflammation while U937 cells are cells derived from the immune system suitable to evaluate the modulation of pro-inflammatory genes. Both cells have been used before and are deemed appropriate to study inflammatory processes in vitro (Zappia, et al., Sci Rep 2015, 5, 17476; Ghosh et al., J Immunol 2010, 185, 3685–3693; Grkovich et al., J Biol Chem 2006, 281, 32978–32987; Nishida et al., Biochem Biophys Res Commun 1988, 156, 269–274; Schreiber et al., Proc Natl Acad Sci U S A 2006, 103, 5899–5904; Sharif et al., BMC Immunol 2007, 8, 1). For their part, MC3T3-E1 pre-osteoblastic cells are widely used to study the maturation of pre-osteoblastic cells into a matrix mineralizing osteoblast (Czekanska et al., Eur Cell Mater 2012, 24, 1–17). This point has been addressed by adding the required information to the main text. Please refer to page 6 lines 255 and 258, and page 9 line 321.

  • After RNA extraction no DNAse treatment was performed. Are all the primer pairs selected in different exons? If not, a possible amplification of genomic DNA cannot be ruled out.

 We would like to thank the reviewer for the observation. It was an involuntary mistake to avoid mentioning the DNAse I protocol, as this is routinely done in our laboratory (Zappia et al., Pharmacol Res Perspect 2019, 7, e00531). Anyway, as a precautionary measure, all primers were designed in different exons to avoid DNA amplification. Both facts were added to their specific section in the text. Please refer to page 4 lines, 156 to 159, and line 183.

  • A table showing the primers and their characteristics is certainly preferable to a list like the one reported in the paragraph: "RT-PCR and Quantitative real-time PCR ".

 We would like to thank the reviewer as we think their comment has made the information more friendly to the readers. The requested table was added as table S1 as Supplementary Material and eliminated from the main text (please refer to page 4 of the manuscript).

  • The DEX concentration is sometimes reported different in the graphs and in the figure legends (or in the text).

 These errors have now been corrected. We thank the reviewer for their observation. Please refer to legends for figures 1, 3 and 5.

  • Quality of table 1 could be improved.

 Following the reviewer’s suggestion, table 1 has been modified to include the confidence intervals.

Reviewer 2 Report

COMMENTS AND SUGGESTIONS FOR AUTHORS

“Antihistamines potentiate dexamethasone anti-inflammatory effects. Impact on glucocorticoid receptor-mediated expression of inflammation-related genes”

The manuscript by Zappia et al. provides the effects of the co-treatment of several antihistamines on the dexamethasone-induced glucocorticoid receptor trascriptional activity on the expression of several inflammation-related genes.

The authors show that some antihistamines increase GCs’ anti-inflammatory effects. Furtermore, since treatment with GCs has many side effects as in bone metabolism, authors investigated about the co-treatment on the expression of bone metabolism markers.

Moreover these data suggest that effects of both, GCs and antihistamines could contribute to decreasing the GCs dose of administration.

Overall the manuscript is well formulated, i have no reservations about publishing the content. Anyway, i would suggest any minor changes.

Materilas and methods: i suggest to write which agents they used to stimulated cells (line 149).

Results: Figure 1A, in my opinion it would be clearer if the authors insert a legend, or if they removed the distinction between white/black. Furhermore it would bee better to explain the meaning of the letters (as “a”, “b”) above the columns, this explanation would also make the following graphss clearer.

Figure 1B: i suggest that the authors include baseline TNF-α expression data before treatment with DEX.

Figure 2 A, 2 B, 2 C: i would like to ask the authors if, pro-inflammatory gene expression was also reduced following treatment with different concentrations of DEX such as 10^-7M.

Figure 3: i would like to ask the authors if the modulation of the anti-inflammatory gene expression was achieved the same following treatment with different concentrations of both DEX and antihistamines.

Regarding both Figure 2 and Figure 3 i ask if it is possible to see the graphs showing absolute gene expression.

Author Response

Point-by point discussion of the reviewer’s 2 comments:

The manuscript by Zappia et al. provides the effects of the co-treatment of several antihistamines on the dexamethasone-induced glucocorticoid receptor trascriptional activity on the expression of several inflammation-related genes.

The authors show that some antihistamines increase GCs’ anti-inflammatory effects. Furtermore, since treatment with GCs has many side effects as in bone metabolism, authors investigated about the co-treatment on the expression of bone metabolism markers.

Moreover these data suggest that effects of both, GCs and antihistamines could contribute to decreasing the GCs dose of administration.

Overall the manuscript is well formulated, i have no reservations about publishing the content. Anyway, i would suggest any minor changes.

Materilas and methods: i suggest to write which agents they used to stimulated cells (line 149).

We thank the reviewer for their suggestion. The agents used are listed in the Materials section along with the suppliers (please refer to page 3 line 113). For the sake of clarity, we have now added the ligands used to the cell culture subsection (page 4 lines 127 to 140).

Results: Figure 1A, in my opinion it would be clearer if the authors insert a legend, or if they removed the distinction between white/black. Furhermore it would bee better to explain the meaning of the letters (as “a”, “b”) above the columns, this explanation would also make the following graphss clearer.

Following the reviewer’s suggestion, we inserted a colour reference in the legend to Figure 1A. We have also rephrased the explanation of the meaning of the letters above each bar, i.e., the letters above bars indicate the result of the statistical test. Bars with a common letter are not significantly different. Likewise, bars not sharing any letter are significantly different. This was detailed in the methodology section. Please refer to page 5, lines 222 to 225.

Figure 1B: i suggest that the authors include baseline TNF-α expression data before treatment with DEX.

Figure 1B is only a transformation of the data shown in Figure 1A. Absolute values of all treatments are depicted in Figure 1A, including the effect of TNF-α on basal levels. Data shown in Figure 1B is a transformation that reflects how each indicated antihistamine modify DEX response (expressed as fold respect to DEX). Using this transformation, the TNF baseline would be equal to zero, since no DEX treatment is present. Hence, Figure 1B shows the effect of the different co-treatments (antihistamine + glucocorticoid) relative to the treatment with dexamethasone alone on the TNF-α-induced gene expression levels.

Figure 2 A, 2 B, 2 C: i would like to ask the authors if, pro-inflammatory gene expression was also reduced following treatment with different concentrations of DEX such as 10^-7M.

To address this question, we show now that increasing DEX concentrations to 10 nM was already sufficient to diminish TNF- or PMA-stimulated pro-inflammatory gene expression, in some cases, below basal levels, preventing the effects of antihistamine. These results are shown in a new Supplementary Section (as figure S1 and S2) and discussed in the main text. Please refer to page 6 line 267.

Figure 3: i would like to ask the authors if the modulation of the anti-inflammatory gene expression was achieved the same following treatment with different concentrations of both DEX and antihistamines.

Dexamethasone concentrations were chosen in accordance with their effect on gene expression. As discussed for figure 2, for GILZ expression in A549 cells, increasing DEX amounts resulted in a much higher effect that could prevent the potentiation induced by antihistamines. For other genes, and in U937 cells, 10 nM DEX induced expression to a level that could be modified by antihistamines. This information was added as figures S3 and S4 of Supplementary Materials and on page 7 lines 296 to 300 of the main text.

Regarding both Figure 2 and Figure 3 i ask if it is possible to see the graphs showing absolute gene expression.

We used the double delta CT method to analyze the qPCR samples. We understand that there are no absolute gene expression values considered in this method since they are all normalized. We prefer this method because it works better when having relatively few DNA samples but many genes to test. We express our results as the fold change of the gene of interest in the test condition, relative to the control condition, which has all been normalized to the housekeeping gene. A fold change of 1 means that there is 100% as much gene expression in the test condition as in the control condition – so there is no change between the experimental group and the control group. A fold-change value above 1 is showing upregulation of the gene of interest relative to the. Values below 1 are indicative of gene downregulation relative to the control. To clarify this methodological point to the reader, we have now included a more detailed description in the Methodology section. Please refer to page 4 lines 188 to 207.

Round 2

Reviewer 1 Report

The authors have paid attention to almost all points raised during the first revision process and replied accordingly, correcting/implementing the text or explaining the reasons for their choices.

I think that the manuscript  should be accepted.